# The Relationships between Screen Use and Health Indicators among Infants, Toddlers, and Preschoolers: A Meta-Analysis and Systematic Review

**DOI:** 10.3390/ijerph17197324

**Published:** 2020-10-07

**Authors:** Chao Li, Gang Cheng, Tingting Sha, Wenwei Cheng, Yan Yan

**Affiliations:** 1Department of Epidemiology and Medical Statistics, Xiangya School of Public Health, Central South University, Changsha 410078, China; Jenniferchaoli@csu.edu.cn (C.L.); gangcheng@csu.edu.cn (G.C.); 2Hunan Key Laboratory of Joint Degeneration and Injury, Xiangya Hospital, Central South University, Changsha 410008, China; tingtingsha@csu.edu.cn; 3Third Xiangya Hospital of Central South University, Changsha 410000, China; wenweicheng@csu.edu.cn

**Keywords:** screen media use, screen-based sedentary behaviors, health indicators, infants, toddlers, preschoolers

## Abstract

Evidence suggests that excessive screen time in early childhood is related to children’s physical and mental health. This study aimed to review the relationships between screen media use and several health indicators in infants, toddlers, and preschoolers. A systematic search was conducted by two independent reviewers on PubMed, Web of Science, Embase, and Cochrane Library to identify the eligible studies, with an end date of 13 August 2019. Included studies (published in English) were peer-reviewed and met the determinate population (children aged 0–7 years with screen media exposure and related health outcomes). The AHRQ, NOS, and the Cochrane Handbook were used to evaluate the cross-sectional study, cohort study, and RCT, respectively. A meta-analysis and narrative syntheses were employed separately. Eighty studies (23 studies for meta-analysis) met the inclusion criteria for the systematic review. Strong evidence of the meta-analysis suggested that excessive screen time was associated with overweight/obesity and shorter sleep duration among toddlers and preschoolers. Excessive screen use was associated with various health indicators in physical, behavioral, and psychosocial aspects. Better-quality research on newer media devices, on various kinds of contents in young children, and on dose–response relationships between excessive screen use and health indicators are needed to update recommendations of screen use.

## 1. Introduction

Children are constantly exposed to media in this digital age [1,2]. As an integrated part of children’s life, screen media use has changed the way children live [1]. Media use has been a popular leisure-time activity [2] and a learning tool [3] for children nowadays. Although many countries and institutes including the American Academy of Pediatrics (AAP) and World Health Organization (WHO) have provided some recommendations [4,5,6] that children aged 0–1 should avoid being exposed to media use, and that children 2–5 years of age were suggested to have no more than two (WHO: <1) hours media use per day to help young children use media devices appropriately, children worldwide do not meet the recommendations [7], having excessive screen media use. According to the recommendation of WHO, excessive screen media use was defined as follows: children aged 0–1 exposed to media devices or children aged 2–7 exposed to screen media use more than 1 h/day. Children use media devices at a very young age [8,9,10], and the frequency and duration of children’s screen time were more than what recommendations suggested [1]. For example, 14.2% of Chinese children [11] aged 3–7 exceed two hours of screen time on weekdays, and 26.7% exceed 2 h/day on weekends, and 19.6% of American children at the age of 2 spend more than 2 h/day on-screen [12]. Since children’s excessive screen media use has become an important public health issue [13], researchers should pay attention to the impacts of excessive screen media use on children’s physical and mental health.

Early childhood plays a vital role in children’s development and health [14]. Evidence suggested that excessive screen media use could have positive [15,16,17] or negative effects [18,19,20] on young children’s health. Excessive screen media use could have positive effects including reducing the attention problem [21], increasing letter knowledge [22], and training learning abilities [3,23]. Excessive screen media use could have detrimental health effects, including overweight/obesity [24,25], short sleep duration [26] and unconsolidated sleep [27], language delay [28], and children’s injury [29]. A growing body of evidence that health outcomes were associated with screen time among young children has been reported in recent years. However, there is no such review that has reported associations between screen time and health outcomes among infants, toddlers, and preschoolers.

Previous reviews focused on factors [30,31,32] of screen use among children and the intervention [33,34,35] to reduce children’s television viewing time. Findings from a systematic review reported 36 factors of children’s screen use, including age, parental media use, family income, parental age, and parental education background [30]. A meta-analysis provided robust evidence that the limitations of children’s screen media use are significantly linked to children’s screen time reduction [35]. Although there have been many studies conducted on the associations between screen time and various health indicators (cognitive development [36], psychosocial effects [37], and children’s weight [38]) among children separately [2], few reviews showed a whole picture of the associations between a broad range of health indicators and screen media use. There was a systematic review that reported the impacts of screen viewing among adolescent girls [39]. However, no review is available for children aged 0–7. Besides, developmental differences in children aged 0–7 years versus adolescence are significant. It is important to examine the younger age group since the development and growth that occurs in early childhood had long-term effects, for example, obesity in early childhood related to grown-up obesity [40]. 

Existing individual studies only focused on the associations between screen time and one health outcome. A summary of the associations between screen time and health outcomes of young children is useful for disseminating and improving screen time recommendations for children aged 0–7, reducing the risk of excessive media use among young children, and identifying future research needs. Therefore, the objective of this work is to provide a global picture of how to relate screen media use with several health indicators in infants, toddlers, and preschoolers. 

## 2. Method

This study was conducted following the Preferred Reporting Items for Systematic Reviews and Meta-Analyses (PRISMA). Detailed statistical procedures used in this systematic review and meta-analysis are shown in Figure 1.

### 2.1. Information Source and Search Strategy 

A systematic search was conducted on PubMed, Web of Science, Embase, and Cochrane Library to identify the eligible studies, published from 1 January 2000 to 13 August 2019. The search strategy was combined with the following strings: (“preschool children” OR “infant” OR “toddler” OR “children”) AND (“social media” OR “media” OR “media exposure” OR “television” OR “television viewing” OR “TV programming” OR “computers” OR “digital computer” OR “electronic games” OR “video games” OR “video” OR “electronic devices” OR “electronic product” OR “digital” OR “mobile” OR “phones” OR “tablets” OR “screen”). 

### 2.2. Study Selection

Two independent researchers (C.L. and T.T.S.) independently screened each record by title and abstract from peer-reviewed journals published from 1 January 2000 to 13 August 2019. The inclusion criteria were as follows: (1) participants: infants, toddlers, and preschoolers (children go to primary school at different ages in different countries, so we defined children’s age as 0–7 years), (2) exposure: durations, prevalence, or viewing categories of screen-based activities, (3) outcomes: physical, behavioral, and psychosocial outcomes related to media exposure, and (4) language: English published articles.

Studies were excluded if they (1) were duplicates, (2) were conference abstracts, reviews, government reports, letters, and protocols, (3) only focused on school-aged children or included preschoolers and school-aged children in one study, or (4) did not report relevant outcomes.

### 2.3. Data Extraction 

Two independent researchers (C.L. and T.T.S.) screened the full texts of the selected studies and extracted data from them. The third independent reviewer (W.W.C.) participated in the discussion if met with any disagreement or discrepancies. Standardized data extracted from eligible studies: author, year of publication, country, sample size, age of participants, media type and measurements of media exposure, and health indicators (e.g., adiposity and sleep problems). The standardized data included details in Appendix A.

### 2.4. Quality and Risk of Bias Assessment

The risk of bias assessment was conducted by two researchers (C.L. and T.T.S.) independently. The evaluation criteria for an observational study of the Agency for Healthcare Research and Quality (AHRQ) [41] were used to evaluate the cross-sectional study, and the Newcastle–Ottawa Quality Assessment Scale for Case–Control Studies (NOS) [42] was used for the cohort study and case–control study. NOS has 8 items with a full score of nine. When a study gets more than six (scores), it is regarded as high quality. When a study gets 4–6 scores, it is regarded as moderate quality. AHRQ is composed of 11 items. Every item of AHRQ was answered as yes, no, or not reported, and only the answer “yes” scored 1, while “no” and “not reported” scored 0. The scores of 8–11 were regarded as high quality, and 4–6 as moderate quality. The Cochrane Handbook evaluated the randomized control studies at outcome levels. Disagreements were settled by discussion among three researchers (C.L., T.T.S., and W.W.C.). Only studies with low risks of bias and high or moderate qualities (without high risks of bias) could be included in this study.

### 2.5. Synthesis of Results

If the included studies were sufficiently homogenous in terms of statistical characteristics, a meta-analysis was planned. Narrative syntheses were planned if they are not sufficiently homogeneous. 

Studies with homogenous statistical characteristics elucidated associations between screen time and physical health indicators (sleep durations (one subsample of sleeping problems) and overweight), so we conducted a meta-analysis based on these studies. We described the characteristics of studies included in the systematic review and conducted a GRADE [43] assessment of the overall quality of evidence of this study. A summary of evidence included in this systematic review was conducted following one model used by numerous reviews [30,44,45]. The details of the coding of the model are described as follows. 

#### 2.5.1. Coding of Health Indicators

Based on the result of team discussion, health indicators associated with screen-based activities were classified as follows: physical outcomes (adiposity, cognitive development, executive function, motor development, musculoskeletal risk), behavioral outcomes (healthy dietary behavior, physical activity, sedentary activity, sleep problems), and psychosocial outcomes (aggressive behavior, behavioral and emotional outcomes, bullying, social and emotional skills, parental interaction (responsiveness)).

The coding of results followed one model used by numerous reviews [30,44,45]. In this model, the relationships between screen use and health outcomes were determined by the total number of studies (Table 1). The findings were coded as follows. When 0–33% of studies reported negative or positive associations between screen time and one of the health outcomes, the association between screen time and the health outcome was defined as no association (NA) (0–33%) in our study. The association between screen time and one of the health outcomes was defined as an unclear association (U) (34–59%) in our study when 34–59% of studies reported positive or negative associations between screen time and the health outcome. When 60–100% of studies reported positive or negative associations between screen use and one of the health indicators, the association between screen time and the health outcome was defined as a positive (+) and negative association (-) (60–100%) in our study. Given that the minimal number of studies could support the individual associations between screen-based activities and health indicators, four or more studies were needed for an overall association, and the results were coded as ++, -- or NANA.

#### 2.5.2. Statistical Analysis 

A meta-analysis was conducted to evaluate the association between overweight/obesity (sleep duration) and media screen use. According to the difference in the recommendations of AAP (<2) and WHO (<2), we defined subsamples as daily screen time ≥1 h/d and ≥2 h/d to extract data and conducted a meta-analysis. We added a subsample as having daily screen time according to screen use in infants and toddlers. Differences in the effect size of the daily screen time ≥ 1 h/d subsample versus data from the daily screen time > 1 h/d subsample were significant. Therefore, we conducted a meta-analysis on the daily screen time ≥1 h/d subsample and daily screen time > 1 h/d when met with minimal observations. As the previous meta-analysis reported, [30] 3 observations in each health consequence or in each exceeding screen time usage were requested, and unadjusted correlation and regression coefficients were abstracted from these analyses. 

The effect size was calculated by using random effects models which were based on the Der Simonian and Laird method, with the Cohen d index. Heterogeneity was assessed by the I^2^ statistic, and its values were considered as not important if I^2^ < 80%, and we also considered the corresponding *p* values.

Publication bias was assessed by conducting the Egger regression asymmetry test, which is significant if *p* < 0.10. Stata software, version 16 (Stata, College Station, TX, USA) was used for statistical analyses. A 2-sided *p* < 0.05 was used to suggest statistical significance.

## 3. Results

### 3.1. Description of Studies

A total of 53,490 records were identified after database searches (Figure 1). There were 39,658 studies remaining after removing the duplicates. Finally, 229 records were needed for full-text assessments. Reasons for excluded records were as follows: could not get full text (*n* = 4), did not report relevant outcomes (*n* = 46), ineligible age (*n* = 56), review studies (*n* = 31), and non-English articles (*n* = 5). Finally, 80 studies, including 37 cross-sectional studies, 36 cohort studies, 1 case–control study, and 6 randomized control studies, were selected in this systematic review. A total of 23 of the 80 studies were included in the meta-analysis.

Characteristics of 80 included studies are provided in Appendix A in Additional Files 1. The age of children who came from 23 countries ranged from 0 to 84 months. Thirty-eight studies were conducted in North American countries, 6 in Oceanian countries, 15 in Asian countries, 18 in European countries, 2 in South American countries, and 1 in an African country. The study involved 209,286 participants in the systematic review, ranging from 6 to 32,439 per study, and involved 49,948 participants in the meta-analysis, ranging from 112 to 13,109 per study.

Regarding the systematic review, 38 studies reported data on physical health indicators, which included adiposity, cognitive development, executive function, motor development, and musculoskeletal risk, 29 studies reported data on behavioral health indicators, which included healthy dietary behavior, physical activity, sedentary behavior, and sleep problems, and 20 studies reported data on psychosocial health outcomes, which involved aggressive behavior, attentional problems, behavioral and emotional outcomes, bullying, social and emotional skills, and parental interaction. Narrative syntheses were conducted since there is a high level of heterogeneity among physical, behavioral, and psychosocial health outcomes, and a meta-analysis was available and conducted only on the physical health indicators: adiposity and sleep problems, separately.

The GRADE assessment is provided in Appendix A. 

### 3.2. Systematic Review

Table 2 shows a summary of all health indicators and their associations with screen time. 

#### 3.2.1. Physical Health Indicators

A total of 22 studies investigated the positive association between screen time and adiposity indicators. Children with excessive screen media use were related to an increased risk of overweight/obesity [24,25,49,50,51,52,53,54,55,56,57,58,59,60,61,62,63,64] and abdominal adiposity [46], increased body mass index (BMI or BMI z-score) [46,47,48,49], sum of skinfold [46,63], and waist circumference [46]. A total of 12 studies examined the associations between screen time and cognitive developments. The results showed that the association between children’s screen time and cognitive development was unclear, waiting for more evidence to figure it out. Children with excessive screen time were associated with an increased risk of delayed language development [28,66,67,69,71,72], language learning problems [23,60,73], mathematics learning problems [23,68,69], and reading problems [68,70]. A total of four studies reported a negative association between screen time and motor developments, and children being exposed to excessive screen time were more likely to suffer from fine motor [73] development problems and gross motor [23,73] development problems. Children having more screen time were associated with worse executive function development, worse motor development, and increased musculoskeletal risk [77,78].

#### 3.2.2. Behavioral Health Indicator

The majority of studies (23/24) narrated a positive association between screen time and sleep problems [19,25,26,27,29,56,70,81,82,83,84,85,86,87,88,89,90,91,92,93,94,95,96]. Children being exposed to screen media use had a higher likelihood of having sleeping problems. Children exposed to excessive screen time were more likely to develop unhealthy dietary behavior [25,48,79,80], and have more sedentary activities [48,77] and insufficient physical activity [25].

#### 3.2.3. Psychosocial Health Indicators

A total of five studies showed the association between screen time and aggressive behavior, and four of these studies reported a positive association, while one study reported no association. Children who had more screen time would have a higher likelihood of being aggressive. Screen time was negatively associated with behavioral and emotional outcomes in 14 studies. Children with excessive screen time were related to social-emotional delay [100,106], hyperactivity-inattention [100,101,105,107,108], emotional symptoms [99,101,102,103], prosocial behavior [99,101,102,103], peer problems [99,101,102,103], and conduct problems [99,101,102,103]. The positive association between screen time and bullying including being a victim or being a bully was reported by three studies [23,109,110]. Children who spend more than the recommended amount of time using media devices would have a higher likelihood of bullying in kindergarten and school. Two studies [21,113] examined the positive relationship between screen time and parent interaction, such as responsiveness, which means that children with excessive screen time were associated with higher responsiveness to their parents.

### 3.3. Meta-Analysis

A total of 23 studies were included in this meta-analysis. The meta-analysis was conducted to explore two associations: the association between screen time and overweight/obesity, and the association between screen time and sleep duration. 

#### 3.3.1. Overweight/Obesity

A total of 9 studies included in the meta-analysis were conducted on the association between screen time and overweight/obesity. The pooled effect size OR estimates for the association between excessive screen time and overweight/obesity are described in Table 3 and Figure 2.

Regarding overweight/obesity, the pooled effect size for exceeding ≥ 1 h/d was 1.872 (95%CI 1.678 to 2.088) (reported by four studies (six items)), for 2 h or more a day it was 1.262 (95%CI 1.155 to 1.379) (reported by four studies (10 items)), and for more than one hour a day it was 1.988 (95%CI 1.445 to 2.735) (reported by four studies (six items)). The heterogeneity *I*^2^ was 21%, 71.8%, and 0 when screen time exceeded ≥ 1 h/d (*p* < 0.001), ≥2 h/d (*p* < 0.001), and >1 h/d (*p* < 0.001), respectively. Children watching an excessive amount of TV or engaging in other screen time activities will have up to twice the risk of obesity.

This subsample was included in the sensitivity analysis. The heterogeneity of the subsample (exceeding ≥ 2 h/d) from Table 3 was 71.8%, and sensitivity analyses indicated that one study [52] conducted on Ethiopia, which had SES (Socioeconomic Status) differences from other studies included in this subsample, may be the reason for the heterogeneity. The Egger asymmetry test suggested that there is no statistically significant publication bias in this meta-analysis.

#### 3.3.2. Sleep Duration

A total of 14 studies were included in the meta-analysis conducted on the association between screen time and sleep duration. The pooled effect size OR estimates for the association between excessive screen time and sleep duration are described in Table 4 and Figure 3.

In the analysis of the association between excessive screen time and sleep duration, four studies (11 items) were included in the group where the duration of media use exceeded 1 h per day, and the pooled effect size was 1.420 (95%CI, 1.392 to 1.449). Regarding sleep duration, the pooled effect size for exceeding ≥ 1 h/d was 2.283 (95%CI, 2.132 to 2.445) (reported by two studies (10 items)), and for two hours or more a day it was 1.053 (95%CI, 1.024 to 1.082) (reported by nine studies (14 items)). The heterogeneity *I*^2^ was 86.8%, 57.6%, and 85.7% when subsamples were defined as having screen time (*p* < 0.001), screen time ≥1 h/d (*p* = 0.009), and ≥2 h/d (*p* < 0.001). Children with excessive screen media use would have more than twice the risk of shorter sleep duration.

The subsample was included in the sensitivity analysis. The heterogeneity of the subsample (exceeding ≥ 1 h/d) from Table 4 was because of the SES described by Marinelli et al. [88]. According to the result of the sensitivity analysis, the SES of Marinelli et al. [88] being different from other studies included in this subsample may be the reason for this heterogeneity. Sensitivity analyses indicated that pooled effect size estimates for exceeding ≥ 2 h and exceeding(Y) in Table 4 were slightly changed when data from these studies were removed. The Egger asymmetry test suggested that there is no statistically significant publication bias in this meta-analysis.

## 4. Discussion

This systematic review and meta-analysis summarizes 80 studies about the relationships between screen time and health indicators among infants, toddlers, and preschoolers. These studies found that more screen media use was related to higher adiposity, more sleep problems and aggressive behaviors, more risk of musculoskeletal pain and bullying in the following years, poorer healthy dietary behavior, worse executive function and motor development, less physical activities and more sedentary activities, and worse behavioral and emotional outcomes. However, the evidence was unclear for cognitive development and the development of emotional and social skills. The meta-analysis elucidates a positive association between excessive screen time and overweight/obesity and a negative association between excessive screen time and sleep duration. 

A review [39] indicated that health indicators were associated with screen-based sedentary behavior among adolescent girls. There is no such review that systematically reported health indicators related to screen time among infants, toddlers, and preschoolers. 

The positive association between screen time and adiposity was elucidated by 10 cross-sectional studies, 10 cohort studies, and 1 case–control study. The meta-analysis suggested that excessive screen time (exceeding ≥ 1 h/d or ≥ 2 h/d) was associated with overweight/obesity among toddlers and preschoolers. Excessive media exposure in early childhood was related to obesity, such as childhood overweight/ obesity [24,46], and obesity in adult age [53]. Screen time also has a relationship with some behavioral health indicators, which were associated with overweight/obesity [48,114]. According to the 24 h theory from Rossano et al. [1], the more screen media time a child had in one day, the less time they left for physical activities since media uses were the main sedentary activities for young children. Screen media use, especially TV watching, along with snacks or fast food intaking, increased the risks of children’s overweight [114]. Meanwhile, most advertisements on television were about sugar, sweet drinks, and other kinds of energy-dense and nutrient-poor food, which could increase the consumption of unhealthy food for young children [79]. This leads to unhealthy dietary behavior for children, as the results of this study showed that screen time was negatively associated with healthy dietary behavior [80]. Besides, evidence suggested that family factors such as SES were relevant to children’s overweight [115], and higher TV time potentially mediated the relationship between childhood overweight and SES [116]. SES of a family was the foundation of consumption of unhealthy food that children watched on TV [116]. Children with low SES were more likely to buy unhealthy food when they had more screen time with sugar advertisements, which could increase the risk of overweight.

Sleeping problems were the indicator with the greatest number of studies (*n* = 23) in this systematic review, and children who had excessive screen time would have a higher likelihood of having sleeping problems. The meta-analysis indicated that excessive screen time was associated with shorter sleep duration in early childhood. Twenge et al. in 2019 [85] suggested that shorter sleep duration correlates with increased use of screen media, and, uniquely, portable devices use strongly associates with shorter sleep duration among children under 10 years old. The mechanisms [26] may be: bluelight from screen media devices that suppressed melatonin levels of children, which changed the circadian rhythm of children; screen media use took up potential sleep duration; screen media use increased physical and psychological arousal; and media exposure improved levels of night alertness and reduced duration of rapid eye movement sleep. Sleep duration decreased with increased age, and Beyens et al. in 2019 [19] pointed out that this is partly because the duration of media use took up sleep duration among children and adolescents. The limitation of screen media use is more important in young children than adolescents. Small screen use (for example, iPod usage and smartphone usage) was more relevant to lack of sleep than large screen use (for example, television watching and tablet playing) [19]. Parents should set limitations on the duration and prevalence of screen media use of infants, toddlers, and preschoolers to meet the recommendations.

The evidence of cognitive development as a health indicator was mixed. On the one hand, media exposure would be beneficial for young children’s word learning [65] since media offers a wide variety of information and fast communication. Target video training would improve the ability of mathematics and reading skills for preschoolers. On the other hand, the magnetic resonance imaging (MRI) evidence in 2019 showed that media use endangered the development of the brain structure by affecting the microstructural integrity of white matter for preschoolers [117]. Screen-based activities such as watching TV and videos for fun were the main part of screen media use among young children. Watching television, especially comic or entertainment, had detrimental effects on children’s cognitive development and behavioral and emotional outcomes [100]. Violent or inappropriate media content for young children was associated with aggressive behaviors and attention problems [97], and even led to bullying when these children went to school [109]. Screen based-sedentary activities, including watching TV or videos, playing computer games, and using a touchscreen, harmed young children’s motor development [73]. However, interactive media use was reported to be positively related to parent interaction, such as the responsiveness of children [21]. The recommendation of screen use should include suggestions on both the content of media exposure and the type of media devices since they are as important as the duration of screen use on children’s development and health. High-quality media use, such as target media training and educational videos, should be an essential tool for young children to improve their learning skills [74], and interactive media use should be used as a tool for increasing parent–child activities and improving parent–child relationships [21]. 

Findings from this systematic review and meta-analysis confirm that excessive screen time, mainly engaging in more than 2 h of daily screen time, has detrimental health effects in both the long and short term. Therefore, children should meet the recommendation of AAP about daily screen media use to avoid suffering from those harmful health effects. Meanwhile, various content of screen use and types of media devices made it complicated to limit the use of media use on children. The recommendation of daily screen use for infants, toddlers, and preschoolers should be improved by conducting more studies about being exposed to new types and contents of media screen use in this digital age.

### 4.1. Strengths and Limitations

This systematic review and meta-analysis has some advantages. First, the inclusions of health indicators related to screen time were comprehensive and accurate. Second, none of the included studies in this systematic review and meta-analysis were at high risk of bias. However, there are some limitations to this systematic review and meta-analysis. First, the duration of children’s screen use was mostly reported by parents, which was at risk of recall bias. Second, the inclusion of only articles published in English could result in English language bias. Third, the number of articles for each indicator was different, which may weaken the credibility of the evidence of health indicators and make an unequal contribution of health indicators related to children’s health. Fourth, the results of this study combined infants, toddlers, and preschoolers, which could not give suggestions on children’s screen use at different ages.

### 4.2. Implications and Recommendations

To reduce the risk of harmful health effects, young children should be encouraged to meet the screen time recommendations of AAP, such as decreasing the duration and prevalence of screen media use, improving the quality and content of media exposure, for example, having targeted media use or interactive media use with parents, and replacing violent contents with educational contents. 

To update the screen time recommendation, more and better studies should be conducted on newer portable media devices and different contents of screen use, and the measurements of screen-based activities should be improved and uniformed as a credible standard.

## 5. Conclusions

This systematic review and meta-analysis was the first to examine the association between screen time and a broad range of health indicators in infants, toddlers, and preschoolers. We provide support for evidence based on original articles showing that excessive screen use was associated with physical, behavioral, and psychosocial health indicators. Strong evidence of this meta-analysis suggested that excessive screen time was associated with overweight/obesity and shorter sleep duration among toddlers and preschoolers. Excessive screen use was associated with various health indicators in physical, behavioral, and psychosocial aspects. The evidence related to cognitive development and the development of emotional and social skills was inconclusive. However, the data from included studies rely on parental-reported screen media use and combined infants, toddlers, and preschoolers. The measurements of screen media use should be improved, and more studies reporting the dose–response relationship should be conducted to better inform screen time recommendation all over the world. Robust screen time guidelines should be based on the expert consensus and best available evidence based on children’s daily life for each country, feasibility, resource use (a common type of media devices for each country), and equity. It is essential for more and better studies of screen media use in newer media devices and various kinds of contents in the young age group, which is significant for children’s growth and development.

## Figures and Tables

**Figure 1 ijerph-17-07324-f001:**
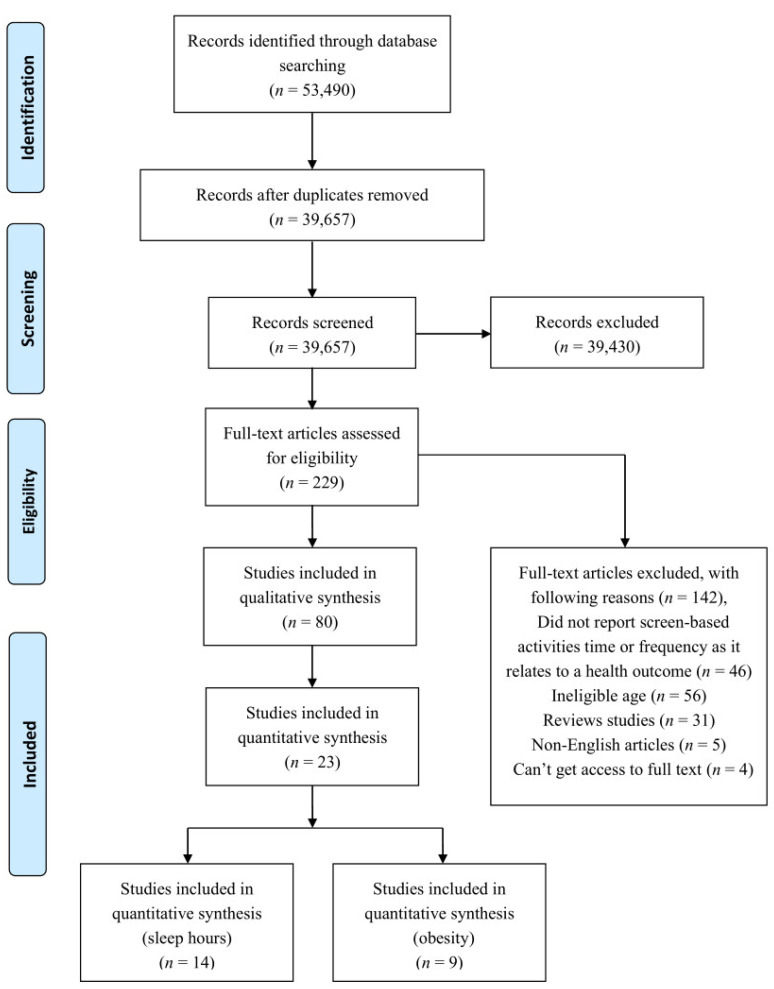
Flow diagram showing the detailed selection of studies included in the systematic review and meta-analysis.

**Figure 2 ijerph-17-07324-f002:**
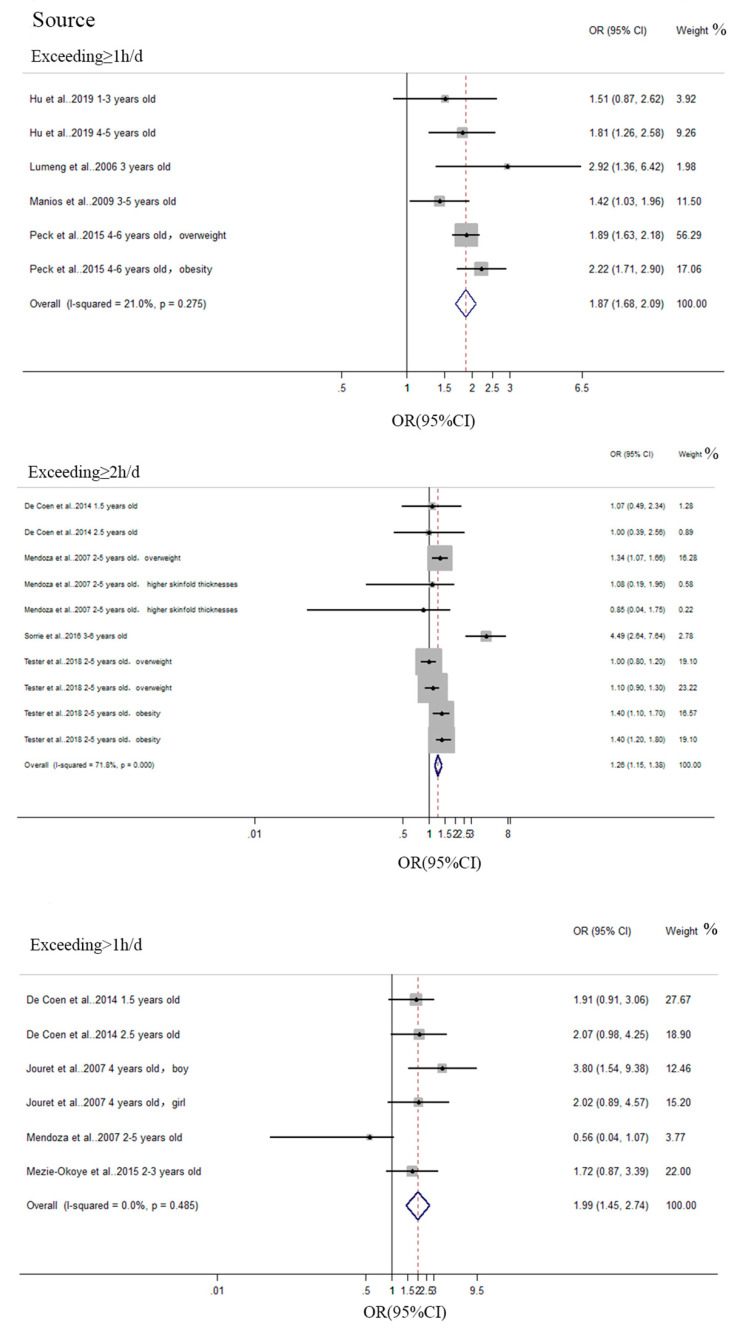
Pooled estimated effect size (ES) of the association between overall screen media time and overweight/obesity.

**Figure 3 ijerph-17-07324-f003:**
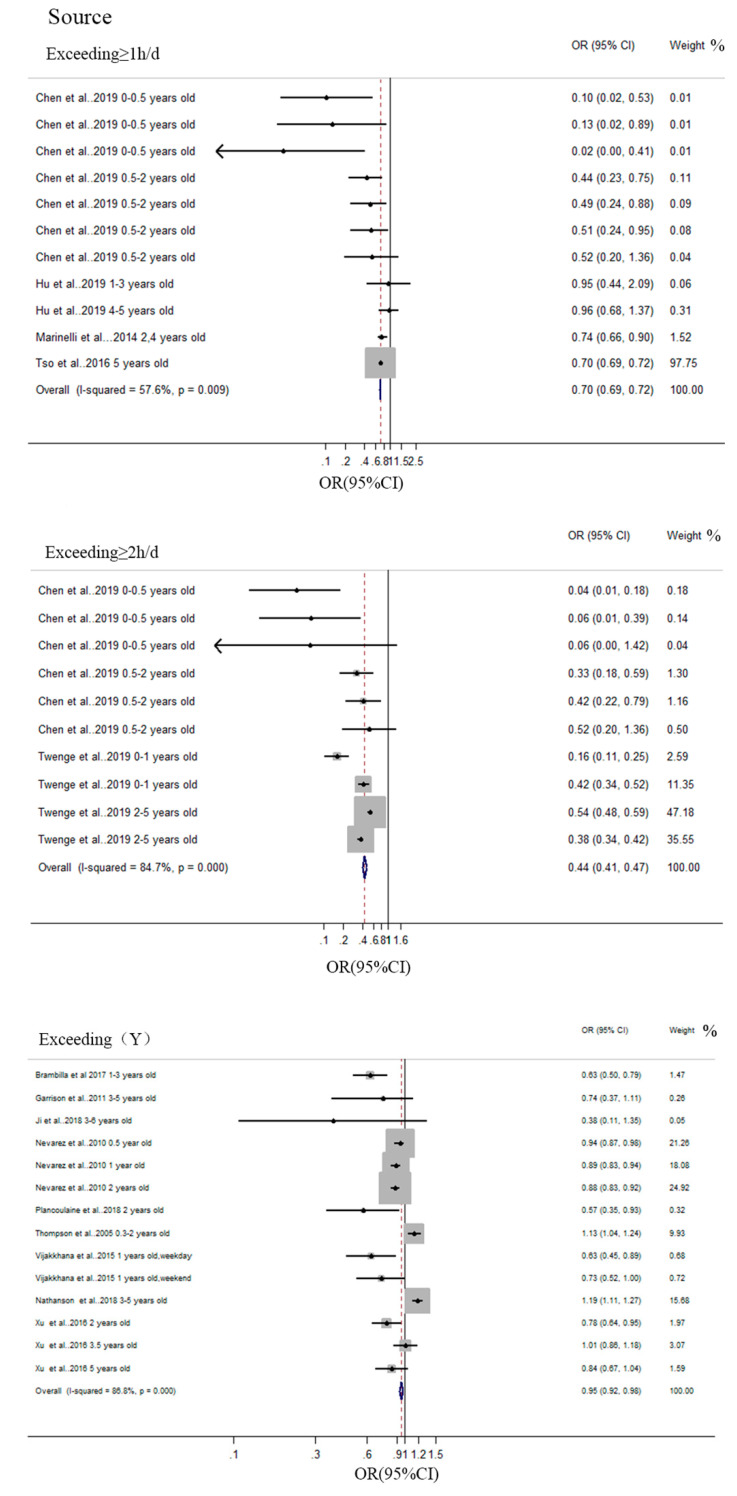
Pooled estimated effect size (ES) of the association between overall screen media time and sleep duration.

**Table 1 ijerph-17-07324-t001:** Rules for classifying associations between screen media exposure and health indicators.

Summary Code	Studies Supporting Association (%)	Meaning of Code
(+)	60–100	Positive association
(-)	60–100	Negative association
NA	0–33	No association
U	34–59	Unclear association

Note: four or more studies support the association/non-association. (+): positive association (60% to 100% of low risks of bias of studies reported positive associations). (-): negative association (60% to 100% of low risks of bias of studies reported negative associations). NA: 0 to 30% of low risks of bias of studies reported positive/negative associations.

**Table 2 ijerph-17-07324-t002:** High-level summary of findings by health indicators.

Variable Classifications	Variables	Positive Association	Negative Association	No Association	Association (% of Studies Examining the Variable that Support Association)
Physical health outcomes	Adiposity	[24,25,46,47,48,49,50], [51] ^(boy)^, [52,53,54,55,56,57,58,59,60,61,62,63]		[64]	(++) 20/21=95.2%
	Cognitive development	[65]	[23,28,66,67,68,69,70]	[69,71,72,73]	(-) 7/12 = 58.3%, (+) 1/12 = 8.3%, NA4/12 = 33.3%, so, unclear
	Executive function		[69,74,75,76]		(--) 4/4 = 100%
	Motor development		[23,67,73]	[73]	(--) 3/4 = 75%
	Musculoskeletal risk	[77,78]			(+) 100%
Behavioral health outcomes	Healthy dietary behavior		[25,48,79,80]		(--) 4/4 = 100%
	Physical activity		[25]		(-)
	Sedentary activity	[48,77]			(+)
	Sleep problems	[19,25], [26] ^(touchscreen, TV)^, [27,29,56,70,81,82,83,84,85,86,87,88,89,90,91,92,93,94,95,96]		[19]	(++) 23/24 = 95.8%
Psychosocial health outcomes	Aggressive behavior	[97] ^(antisocial)^, [98,99]			(++) 3/3 = 100%
	Behavioral and emotional outcomes	[100] ^(Prosocial behavior)^	[11,75,93,99], [100] ^(Hyperactivity)^, [101,102,103,104,105,106,107] ^(Attentional problem)^, [108] ^(Attentional problem)^	[100] ^(Peer problems, Emotional symptoms, Conduct problems)^	(--) 12/14 = 85.6%
	Bullying	[23,109,110]			(+)
	Social and emotional skills	[111]	[112]		unclear
	Parental interaction (Responsiveness)	[21,113]			(+)

Notes: (+) = 60% to 100% of low risks of bias of studies reported positive associations and 3 or fewer studies reported associations between screen time and the health indicator. (++) = 60% to 100% of low risks of bias of studies reported positive associations and 4 or more studies reported associations between screen time and the health indicator. (−) = 60% to 100% of low risks of bias of studies reported positive associations and 3 or fewer studies reported associations between screen time and the health indicator. (--) = 60% to 100% of low risks of bias of studies reported negative associations and 4 or more studies reported associations between screen time and the health indicator. NA = studies consistently report no association.

**Table 3 ijerph-17-07324-t003:** Summary of the meta-analysis of the association between media screen use and overweight/obesity.

Subsample	No. of Studies	OR (95%CI), Random Effect Models	P_Q_	I^2^ (%)	Egger Result
Exceeding ≥ 1 h/d	4 (6)	1.872 (1.678,2.088)	*p* < 0.001	21%	*p* = 0.943
Exceeding ≥ 2 h/d	4 (10)	1.262 (1.155,1.379)	*p* < 0.001	71.80%	*p* = 0.588
Exceeding > 1 h/d	4 (6)	1.988 (1.445,2.735)	*p* < 0.001	0	*p* = 0.456

Notes: the control group of Exceeding ≥ 1h/d was screen time < 1 h/d. The control group of Exceeding ≥ 2 h/d was screen time < 2 h/d. The control group of Exceeding > 1 h/d was screen time ≤ 1 h/d. **P_Q_** < 0.05 is significant. Egger result: *p* < 0.10 is significant.

**Table 4 ijerph-17-07324-t004:** Summary of the meta-analysis of the association between media screen use and sleep duration.

Subsample	No. of Studies	OR (95%CI), Random Effect Models	P_Q_	I^2^ (%)	Egger Result
Exceeding ≥ 1 h/d	4 (11)	1.420 (1.392, 1.449)	*p* = 0.009	57.6%	*p* = 0.102
Exceeding ≥ 2 h/d	2 (10)	2.283 (2.132, 2.445)	*p* < 0.001	84.70%	*p* = 0.069
Exceeding (Y)	9 (14)	1.053 (1.024, 1.082)	*p* < 0.001	86.80%	*p* = 0.146

Notes: the control group of Exceeding ≥ 1 h/d was screen time < 1 h/d. The control group of Exceeding ≥ 2 h/d was screen time < 2 h/d. The control group of having screen time named as Exceeding (Y) was not having screen time. **P_Q_** < 0.05 is significant. Egger result: *p* < 0.10 is significant.

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
