# Peer review of "The Relationships between Screen Use and Health Indicators among Infants, Toddlers, and Preschoolers: A Meta-Analysis and Systematic Review"

_ijerph, 2020, doi:10.3390/ijerph17197324_

Round 1

Reviewer 1 Report

This study was well conducted and clearly described. The researchers point out the limitations (which are always somewhat high in meta-analyses because of the differences in studies). It is an important topic and worthy of publication. The descriptions of the processes used to select studies and the rationale for conclusions are well stated. This study would be of interest to many professionals in the field of early childhood education.

Author Response

Dear reviewer of International Journal of Environmental Research and Public Health,

On behalf of my co-authors, we appreciate editors and reviewers very much for your valuable and constructive comments and suggestions on our manuscript entitled “The relationships between screen use and health indicators among infants, toddlers, and preschoolers: A meta-analysis and systematic review” (IJERPH- 912873). Those comments are very valuable and helpful for revising and improving our paper, as well as the important guiding significance to our researches. We have studied comments carefully and have made correction which we hope meet with approval. A version of our revised manuscript was uploaded and labeled 'Revised Manuscript'.

Thank you and best regards. 

Yours sincerely,

Corresponding author: Yan Yan

Department of Epidemiology and Health Statistics

Xiangya School of Public Health, Central South University

Reviewer 2 Report

Please expand your introduction and conclusion sections but in particular your introduction is too meagre. Rest is fine. You also need to make minor English corrections so it reads more fluently; words are missing at the moment, making it hard to understand what you mean. 

Reviewer 3 Report

I enjoyed reading this well-written paper in a public health area. The aim of this study was to review the relationships between screen media use and several health indicators in infants, toddlers, and preschoolers. I have some minor issues to consider by authors: *Abstract - It is advisable to add “AHRQ” and “NOS” in the abstract. It is important to give information about the methods used to a meta-analysis and systematic review. - Please specify in the abstract how many articles were included in the meta-analysis and how many in the systematic review. *Method - Did you register your study? - Page 3, line 79: please, add also start date. - Page 4, Quality and Risk of Bias Assessment: please, provide references for AHRQ and NOS. Please, provide information where are the results of quality and risk of bias assessment (Table S2?). Please change the name of table S2 to indicate that it includes the results of quality and risk of bias assessment. Table S2 indicates that the ‘quality’ of the included studied was “very low”, however I did not find the scores of quality assessment which were regarded as “low” or “very low”. What would be very low methodological quality? It is important to set this issue into the method. *Results Table 1. Please, explain under the table what “++” and “- - “ means. *Discussion (4.1. Strengths and limitations): You wrote “Second, none of the included studies was at high risk of bias”, but in Table S2 you stated that “quality” of included studies was “very low. Please clarify the information contained in the methods (Quality and Risk of Bias Assessment) and Table S2, to make them more readable.

Reviewer 4 Report

This is an interesting topic in the more digitalized society in which we live, with touchscreens making surfing more and more accesible for children.

  1. I find that the introduction is to brief and partly confusing. You need to look both at language and clarify the bioecological framework. Your aim and research questions are not clear.

  1. It is an impressive work going thru this type of large material. I think however that you should clarify how many of the studies in meta analysis that are cross-sectional Also tables are confusing.
  2. Table 1: what does a positive association mean? What do a negative association mean? Is this in relation to what you are looking for or what the original study was looking for. Is a positive related like higher weight and more screentime while a negative is more screentime less sleep. In that instance it actually both confirm your hypothesis and should not be coded as positive and negative. I do not understand the percentage either, this need to be clarified. I Think this relates to table 2 as well. because looking at this table what does it say. If you would have had a hypothesis and the number of studies confirming your hypothesis and not confirming it would be better. I do not get the ++ - + - here it is not clear in text what you mean (i do get that + in positive association). But it is not clear.
  3. Table 3: what is the differences between exceeding < 1h/d and exceeding <1h/day? You give me the nonsignificance of the eggers test. You do not explain neither in data procedure nor here what I square stands for, or what cut-offs you been using. Looking at for example subgroup Exceeding >2h/day  the I2 is 71,8% which shows serious heterogeneity and no analysis on subsamles should be conducted. But you are perhaps not doing this either.
  4. You are talking about effect size OR. For overweight this is clear, you are more overweight if you watch more TV. However what comes before? In discussion you talk about commercial , and it might be relevant. Family factors such as SES might also be relevant. As we know that weight and SES are relevant.
  5. Looking at table 4. your OR is below 1. indicating a lower risk. But you are talking about sleepproblems as having a positive association with sleep problems. This mean that you need to clarify why the OR is below 1 not above it, what are you measuring. Also in table 4. I am not sure what group Y, that is the control? i.e. not looking at at at screen? In this table the I2 value is exceeding 50% for all groups indicating serious problems with heterogeneity. For example for two of the groups more than 80 percent of the result are due to between study variability.

  1. Appendix: hard to connect appendix S1 and S2, when S2 doesn't have the reference. I suggest that you include the number of the reference in the table, so that it is more reader friendly.

  1. Including cross-sectional studies makes it hard with causality, only associations, which mean that we do not know if people thar have health issues choose to be more sedentary, or if they because of being more sedentary have more health issues.

  1. Language, the syntax feels wrong, perhaps let someone review the language? Also a mixture of tempus in sentences and in paragraphs, makes it hard to follow.

  1. On page two second paragraph, you are talking about the content not the time as positive or negative, which isn't in line with the rest of the reasoning in that paragraph, because you are talking about time, not content as the main issue.

  1. How will you use a bioecological theoretical framework? No reference. I can only see biological aspects in text. What other levels are you measuring, there are missing information here.

Round 2

Reviewer 4 Report

I find the methodology much clearer, and it is easier to follow and understand the results. Which is good. 

I still think that the authors are to close to the statistics and actually only describe what is in the tables, which is redundant. I think that it would be better if they explain what it means for children, be more narrative. they are a bit more explanatory in discussion but still very much associations, relations, very little about children. 

Result section, at least regarding the meta analysis need to be more reader friendly. would a division between the different outcomes, perhaps with a subheading make this easier to follow together with what it implies. 
